# Photoredox-Catalyzed Reduction of Halogenated Arenes in Water by Amphiphilic Polymeric Nanoparticles

**DOI:** 10.3390/molecules26195882

**Published:** 2021-09-28

**Authors:** Fabian Eisenreich, Tom H. R. Kuster, David van Krimpen, Anja R. A. Palmans

**Affiliations:** Laboratory of Macromolecular and Organic Chemistry, Institute for Complex Molecular Systems, Eindhoven University of Technology, P.O. Box 513, 5600 MB Eindhoven, The Netherlands; f.r.eisenreich@tue.nl (F.E.); t.h.r.kuster@student.tue.nl (T.H.R.K.); d.v.krimpen@student.tue.nl (D.v.K.)

**Keywords:** amphiphilic copolymer, photoredox catalysis, polymer assembly, hydrophobic collapse, compartmentalization, dehalogenation

## Abstract

The use of organic photoredox catalysts provides new ways to perform metal-free reactions controlled by light. While these reactions are usually performed in organic media, the application of these catalysts at ambient temperatures in aqueous media is of considerable interest. We here compare the activity of two established organic photoredox catalysts, one based on 10-phenylphenothiazine (**PTH**) and one based on an acridinium dye (**ACR**), in the light-activated dehalogenation of aromatic halides in pure water. Both **PTH** and **ACR** were covalently attached to amphiphilic polymers that are designed to form polymeric nanoparticles with hydrodynamic diameter D*_H_* ranging between 5 and 11 nm in aqueous solution. Due to the hydrophobic side groups that furnish the interior of these nanoparticles after hydrophobic collapse, water-insoluble reagents can gather within the nanoparticles at high local catalyst and substrate concentrations. We evaluated six different amphiphilic polymeric nanoparticles to assess the effect of polymer length, catalyst loading and nature of the catalyst (**PTH** or **ACR**) in the dechlorination of a range of aromatic chlorides. In addition, we investigate the selectivity of both catalysts for reducing different types of aryl-halogen bonds present in one molecule, as well as the activity of the catalysts for C-C cross-coupling reactions. We find that all polymer-based catalysts show high activity for the reduction of electron-poor aromatic compounds. For electron-rich compounds, the **ACR**-based catalyst is more effective than **PTH**. In the selective dehalogenation reactions, the order of bond stability is C-Cl > C-Br > C-I irrespective of the catalyst applied. All in all, both water-compatible systems show good activity in water, with **ACR**-based catalysts being slightly more efficient for more resilient substrates.

## 1. Introduction

Photoredox catalysis has emerged in the last decade as a powerful tool for chemical bond transformations and late-stage functionalization of pharmaceutically relevant compounds [1,2,3,4,5]. Since reactions only occur when the catalyst is in its excited state, which can be accurately controlled by light, photoredox-catalyzed reactions show an unprecedentedly high degree of spatial and temporal control. Traditionally, photoredox-catalyzed reactions have been performed in organic solvents, as this permits good mixing between the catalyst, the substrates and the products. More recently, photoredox catalysis has also become feasible in water by using micellar systems, water compatible ligands, or water-soluble counter ions for the metal-based catalysts [6,7,8,9,10]. Water is considered a green solvent, making reactions more environmentally benign. At the same time, water also facilitates recyclability of the catalysts, which is of high interest when expensive transition metals are applied [6,8].

Organocatalytic alternatives for photoredox catalysts have been developed to avoid the use of expensive Ir- and Ru-based systems and to tune the redox potential of the catalysts [5,11,12,13]. Their synthetic potential has been illustrated by their applicability for a broad spectrum of substrates and in a range of reactions such as polymerizations, substitutions, bond cleavage reactions and cycloadditions [2,14,15,16]. Usually, these organocatalysts are evaluated in organic media, making separation of products, solvent and catalyst laborious. Additionally, here, the use of water as solvent by introducing micellar systems [17,18,19,20,21] or by using water-soluble catalytic complexes [22,23] have shown great potential. Micellar systems are especially interesting since the hydrophobic domains present in the aqueous medium allow us to increase the local catalyst concentration, and when hydrophobic substrates are used this results in faster conversion at equal catalyst loading [24,25]. However, if the catalyst is not covalently bound, it can exit the hydrophobic domains. To prevent the catalyst from leaking out, covalent linking of the catalyst to a polymer backbone is a promising way to retain high catalyst activity and stability [26,27,28]. We recently observed that covalently attaching the highly reductive 10-phenylphenothiazine catalysts to an amphiphilic polymer resulted in stable and highly active photoredox catalysts in water. In addition, we observed that the polymer catalysts could be reused without loss of activity up to five times [23]. 

In this work, we present a systematic investigation on the reactivity of two organic photoredox catalysts namely 10-phenylphenothiazine (**PTH**) [14,29] and the acridinium dye 9-mesityl-3,6-di-*tert*-butyl-10-phenyl-acridinium (**ACR**) [16] covalently attached to water-soluble polymeric nanoparticles (**PNs**) (Figure 1). The **PNs** permit the catalyst to be taken up in the hydrophobic pocket [30,31,32,33], which together with accumulation of hydrophobic substrates affords a high local concentration of both the catalyst and the substrate. The acridinium catalyst (E_1/2_* = −3.36 V vs. SCE) is an exceptionally strong reductant in its excited state but needs to be excited twice before the desired excited state is reached [16]. The **PTH** catalyst (E_1/2_* = −2.1 V vs. SCE) is also a very strong reductant but less potent when compared to the **ACR**. It does, however, require one excitation only to reach the desired excited state [14,15]. For **PTH**, dehalogenation reactions [14,15,23], C-C cross-coupling reactions [15,23] and radical polymerizations [29,34] have been reported. **ACR** catalysts have also been investigated in dehalogenations [35], cross-coupling reactions [36], halogenations [36] and radical polymerizations [37]. We here select the dehalogenation of aryl chlorides as a benchmark reaction to compare the two systems in water and to elucidate how far the degree of polymerization, catalyst loading, and the nature of the catalyst, **PTH** or **ACR**, affect the reactivity towards dechlorination reactions. In addition, we investigate in how far selectivity can be attained in dehalogenations of different carbon–halogen bonds. Finally, we evaluate the feasibility of a cross-coupling reaction of an aryl chloride with *N*-methylpyrrole.

## 2. Methodology and Results

### 2.1. Molecular Design, Synthesis, and Characterization of Photocatalyst Loaded PNs

We designed and synthesized six different amphiphilic polymers, **PN1**–**6**, with the aim of assessing the influence of the polymer chain length, catalyst loading and catalyst type on catalyst activity in aqueous solutions (Figure 2). To this end, we first synthesized two polyacrylates (**P_100_** and **P_200_**) with degrees of polymerization (DP) of 100 and 200, using pentafluorophenyl acrylate as the monomer [38]. The two polyacrylates were made by reversible addition–fragmentation chain-transfer (RAFT) polymerization of the acrylate, followed by the removal of the RAFT end group [39]. The DP was determined by monitoring the monomer conversion with ^19^F NMR spectroscopy. The molar mass distributions (*Đ*) of the polymers were narrow (e.g., *Đ* (**P_200_**) = 1.15) as evidenced by size-exclusion chromatography (SEC), calibrated by polystyrene standard in THF, indicating the controlled nature of the polymerization (Appendix A). 

In order to covalently attach the selected photoredox catalysts, **PTH** and **ACR**, to the polymer backbone, both catalysts were equipped with an amine functional group. **PTH-NH_2_** was prepared according to our previously reported procedure [23] whereas **ACR-NH_2_** was synthesized following a modification of the procedure reported for **ACR** [35]. By changing the aniline used in the final synthesis step to BOC-protected 4-(aminoethyl)aniline, followed by removal of the BOC group, the desired **ACR-NH_2_** was readily accessed. The details of the **ACR-NH_2_** synthesis and its molecular characterization are given in the Appendix A. 

After successfully acquiring both amine-functional catalysts, the reactive ester side groups of the poly(pentafluorophenyl) acrylates were sequentially substituted by the desired **PTH-NH_2_** (5%) or **ACR-NH_2_** (5 or 20%), followed by hydrophobic *n*-dodecylamine (15 or 0%), and hydrophilic Jeffamine@1000 (80%) (Figure 2). We added *n*-dodecylamine as additional hydrophobic moiety as it enhances the hydrophobic collapse of the polymers in water [40]. A random distribution of the hydrophobic/hydrophilic moieties along the polymer chain is important as this helps to induce an intramolecular, hydrophobic collapse of the amphiphilic copolymer in water [41,42], which results in the formation of unimolecular polymeric nanoparticles of defined size. The conversion of each step when adding the amines sequentially was assessed by ^19^F NMR spectroscopy by comparing the sharp peaks of the released pentafluorophenol to the broad peaks of the polymer bound pentafluorophenyl group (Appendix A). We synthesized six different, fully functionalized polymers (**PN1**–**6**). **PNs** with a DP of 100, **PN1**–**3**, have a theoretical molecular weight of 93–101 kDa and *Đ* of 1.15–1.19. **PNs** with a DP of 200, **PN4**–**6**, have a theoretical molecular weight of 187–202 kDa and *Đ* of 1.17–1.31. The molar mass distributions were determined by SEC, calibrated with poly(ethyleneoxide) in DMF with LiBr (full SEC traces are in the Appendix A).

All polymers were well soluble in water. Their hydrodynamic diameters (*D*_H_) in water were determined by dynamic light scattering (DLS) measurements (Appendix A). The DLS traces show unimodal size distributions, with *D*_H_ ranging between 5 and 11 nm. These sizes are in line with those observed for systems of similar DP and microstructure, and indicate that all polymers, including those with 20% acridinium groups attached, form defined nanoparticles in water [39,40]. DLS, however, does not exclude that particles comprise multiple polymer chains. The larger sizes of some of the polymers (**PN1**, **PN2** and **PN5**) may indicate some clustering of polymer chains. For the catalytic studies, however, this is not important, as long as the particles are of defined size. 

Finally, the optical properties of the nanoparticles were studied with UV/Vis spectroscopy with water as solvent (Appendix A). It was previously reported that the **PTH** absorption maxima are located around 258 and 320 nm [43], whereas **ACR** absorption maxima occur at 360 and 520 nm [16]. Both the polymer bound **ACR** and **PTH** moiety show absorption bands identical to those reported in the literature for the free catalysts. This shows that both catalysts have been successfully attached to the **PNs** and that the optical properties are not affected by the presence of the amphiphilic polymer.

### 2.2. Reduction of Para-Substituted Aryl Chlorides in Water Using PNs 

As a benchmark and to compare the reducing efficiency of our **PNs** to previously reported systems in organic media, we selected the dechlorination of aryl chlorides [14,16,23,43,44]. We selected five chlorinated benzene derivatives **S1**–**S5** each with a different para-substituent, ranging from electron-donating (**S2**) to electron-withdrawing (**S1**, **S3**) or only exerting inductive effects (**S4**, **S5**) (Table 1). Regarding the choice of substrates, we did not include substrates with strong electron donors, such as dimethylamine groups, as they could potentially act as the proton or electron source and thus result in side reactions. Nitro compounds, as electron poor substrates, were initially tested but yielded many side products, as nitro substrates undergo photolysis reactions under light illumination [45]. All reactions were performed after degassing the solutions with argon in stirred glass vials, using 385 nm light-emitting diode (LED) light. Note that the presence of a tertiary amine base, such as *N*,*N*-diisopropylethylamine (DIPEA), is paramount for the catalytic cycle as it acts as a sacrificial proton and electron source [14,16]. Details of the reaction setup and the involved reaction mechanisms are given in the Appendix A. The conversions were measured by NMR spectroscopy (Appendix A) and the results of the dechlorinations are shown in Table 1.

Electron poor aromatic systems are expected to show higher conversions compared to electron rich systems [16,23]. The results indeed show that electron poor **S1** is completely reduced after 3 h for all **PN1**–**6**. With **S3**, the conversions are lower, between 23 and 55%. Electron rich **S2**, in contrast, showed a significant decrease in conversion, which only ranged between 2 and 18%. Additionally, methylated and trifluoromethylated **S4** and **S5**, less electron poor than **S1**, show low conversions ranging between 0 and 22%. Interestingly, the 5% **PTH**-loaded polymers **PN1** and **PN4** showed a higher conversion than the 5% **ACR**-loaded polymers **PN2** and **PN5**. When looking at the more electron rich substrates 4-chlorotoluene **S3** and 4-chlorobenzotrifluoride **S4**, the 5% **ACR-**loaded polymers show higher conversion compared to the **PTH**-loaded analogues, indicating the importance of a higher reduction potential for these more resilient substrates. 4-Chloroanisole **S2** showed higher conversion for both the **PTH**-loaded polymers, despite the fact that the methoxy group is an electron donating group. The influence of catalyst loading on conversion turned out as anticipated. The polymeric nanoparticles with 20% **ACR PN3** and **PN6** outperformed the 5% **ACR** polymers **PN2** and **PN5** for every substrate, which is likely related to the decrease in the S/C ratio from 90:1 to 25:1. No obvious correlation between the length of the polymeric backbone and conversion was established, indicating that the DP of the polymer is not relevant to tune catalyst activity.

All in all, **PN3** and **PN6** show the highest conversion for most of the used substrates. The activity of the **PTH**-loaded polymers **PN1** and **PN4** is high for less resilient substrates, but quickly drops in activity as the electron-withdrawing nature of the substituent decreases. Higher conversions can, however, be reached by irradiating the reaction mixtures for longer periods of time. Methyl 4-chlorobenzoate **S3** showed almost full conversion (90–98%) for all six polymers after 18 h of irradiation.

### 2.3. Investigating Selectivity of Dehalogenation with PTH and ACR

As shown above, different conversions for the dechlorinations were observed between **ACR**- and **PTH**-based **PNs** between electron poor and electron rich aromatic chlorides. A difference in conversion can also occur when different types of halogens are attached to the benzene ring, due to the different reduction potentials between C-Cl, C-Br and C-I bonds [14,15,16]. The reduction potential decreases from C-Cl (E_1/2_ = −2.8 V for chlorobenzene) [46] to C-I (E_1/2_ = −1.6 V for iodobenzene) [47] bonds, making C-Cl the most resilient to reductive dehalogenation, and C-I the most reactive [48]. To assess whether the **PNs** in water with either **PTH** or **ACR** attached show selectivity towards reducing carbon–halogen bonds, benzenes with different carbon–halogen bonds were evaluated as substrates. As previously shown by Poelma et al. [15] in organic media, selective dehalogenation can be achieved by controlling the reaction time or using photoredox catalysts with different excited state reduction potentials.

To identify the difference in dehalogenation capacity of both photocatalysts, three substrates were selected to perform these selectivity reactions: 1-chloro-4-iodobenzene **S6**, 1-bromo-4-chlorobenzene **S7** and 2-bromo-4-chloro-1-iodobenzene **S8**. The conversion of these reactions was determined via ^1^H NMR spectroscopy (Appendix A) and the results are summarized in Table 2.

The C-I bond in 1-chloro-4-iodobenzene **S6** was preferentially reduced by all polymers, but some C-Cl was also reduced (between 1–8%). Interestingly, the 20% **ACR**- loaded polymers **PN3** and **PN6** showed the lowest conversion of **S6** but the highest amount of benzene formation, indicating a lower activity and lower selectivity. The 5% **ACR**-loaded polymers **PN2** and **PN5** showed similar conversions to the **PTH**-loaded polymers, but a slightly higher conversion towards benzene. This suggests that **PTH**-based **PNs** are slightly more selective in breaking C-I bonds in halogenated aromatic compounds, likely a result of their lower reducing ability. 

1-Bromo-4-chlorobenzene, **S7**, showed, as expected, significantly lower conversions, ranging from 6 to 49%, a result of the higher reduction potential of the C-Br bond. Especially the **PTH**-loaded polymers **PN1** and **PN4** showed a significant decrease in catalytic activity, indicating the importance of the strong reductive nature of the excited state of acridinium to reduce more resilient halogens. Additionally, here, only acridinium loaded polymers were able to reduce the halogenated substrate to benzene, with the two 20% acridinium loaded polymers, **PN3** and **PN6**, forming the most benzene. 

Finally, 2-bromo-4-chloro-1-iodobenzene, **S8**, was reduced to identify which products are mainly formed. The major products were 1-bromo-3-chlorobenzene **R10** and chlorobenzene **R6**, and neither bromobenzene **R9** nor iodobenzene **R7** were formed, indicating the clear preference for cleavage of carbon halogen bonds for halogens with lower reduction potentials. In none of the examples discussed above did we observe a difference in conversion between polymers with different degrees of polymerization, suggesting that polymer length does not play a notable role in the activity or selectivity of the catalysts.

### 2.4. C-C Cross-Coupling Reaction with N-Methylpyrrole

We explored the C-C cross-coupling reaction between 4-chlorobenzonitrile, **S1**, and *N*-methylpyrrole, **S9**. 4-Chlorobenzonitrile **S1** was selected as most suitable substrate as it showed full conversion after 3 h of irradiation for all six **PNs**. In this reaction, two product formations are competing, namely the formation of C-C cross-coupled **R11** and reduced product **R1** (Table 3, Appendix A). In order to shift the selectivity of this reaction to the C-C cross-coupled product, an excess of *N*-methylpyrrole (25–50 equivalents) is usually added [15,46]. We previously observed in the reaction of 2-iodobenzonitrile with *N*-methylpyrrole that the high local concentration of *N*-methylpyrrole within the hydrophobic compartments of nanoparticles led to an increase in the selectivity towards the C-C cross-coupled product, thus only five equivalents of *N*-methylpyrrole were required [23]. Here, we find a selectivity towards the cross-coupled product **R11** up to 48%. Both the 5% **PTH**- and 20% **ACR**-loaded polymers **PN1**, **PN3**–**4**, and **PN6** showed a similar selectivity of roughly 47% towards **R11**. The 5% **ACR**-loaded polymers **PN2** and **PN5**, surprisingly, showed a lower selectivity towards the cross-coupled product of roughly 35%. In all cases, the selectivity is somewhat lower than that previously reported for PTH-loaded systems, which showed a selectivity for the cross-coupled product of up to 78% using 2-iodobenzonitrile as the substrate. 

## 3. Discussion

The two reductive organic photoredox catalysts, **PTH** and **ACR**, covalently attached to an amphiphilic polymeric backbone show promising results in the reduction of aryl chlorides in aqueous media. It has to be noticed, however, that conversions seem rather low, especially for the more resilient substrates. This is mainly related to the low catalyst concentrations we used (1 or 4 mol%) and the limited reaction time of 3 h. When comparing the results found in this work to results previously reported, which is challenging as the conditions are never identical, we can make some interesting observations. **PTH**-loaded **PNs** are much more efficient in the reduction of 4-chlorobenzonitrile and methyl 4-chlorobenzoate in water than free **PTH** in acetonitrile: Using 5 mol% of catalyst, Discekici et al. needed 72 h to achieve 92% conversion for both substrates [14], while in our case complete conversion was reached within 3 h for 4-chlorobenzonitrile and 16 h for methyl 4-chlorobenzoate using only 1 mol% of catalyst. Additionally, for **ACR**-loaded **PNs** the reduction of 4-chlorobenzonitrile in water is more efficient than in acetonitrile: Nicewicz et al. needed 16 h to achieve 78% conversion using 10 mol% catalysts whereas we found full conversion in 3 h for 1 mol% catalyst [16]. For the more demanding substrates, 4-chloromethoxybenzene and 4-chlorotrifluoromethylbenzene, Nicewicz et al. reported conversions of 82 and 58%, respectively, after 16 h using 10 mol% catalysts, whereas we found best conversions of 18 and 22%, respectively, after 3 h with 4 mol% catalyst [16]. It is likely that similar or even higher conversions will be attained when the reaction runs for 16 h. All in all, the most noticeable difference between the different **PNs** reported in this work, is the lower selectivity that **ACR**-loaded **PNs** show in dehalogenations. Although the differences are not that significant, it is clear that **PTH** is the best catalyst when selectivity is desired. On the other hand, **ACR**-loaded **PNs** are active for more resilient substrates.

## 4. Materials and Methods

### 4.1. Materials

All solvents and chemicals used were of reagent grade quality and purchased from Biosolve (Valkenswaard, The Netherlands) or Sigma-Aldrich (Amsterdam, The Netherlands) at the highest purity available and used without further purification unless otherwise noted. Water for aqueous samples was purified on an EMD Millipore Milli-Q Integral Water Purification System (Merck, Amsterdam, The Netherlands). Methyl 2,4,6-trimethyl benzoate [49], 3,6-di-*tert*-butyl-9-mesitylxanthylium tetrafluoroborate [16], **PTH-NH_2_** [23], and **P_100_** [23] were prepared according to procedures detailed in the literature.

### 4.2. Methods

NMR measurements were performed on a Bruker 400 MHz Ultrashield spectrometer (Karlsruhe, Germany) (^1^H at 400 MHz, ^13^C at 100 MHz and ^19^F at 376 MHz) at 25 °C. The compound of interest was dissolved in CDCl_3_, obtained from Cambridge Isotope Laboratories (Tewksbury, MA, USA), with the internal standard for ^1^H and ^13^C δ_CDCl3_ = 7.26 ppm and δ_CDCl3_ = 77.16 ppm, respectively. Multiplicities are indicated with the following abbreviations: singlet (s), doublet (d), doublet of doublet of doublets (ddd), triplet (t), quartet (q), multiplet (m), and broad (br). The obtained NMR spectra were processed using MestReNova v14.0.1-23559 (Mestrelab Research, Santiago de Compostela, Spain). MALDI-TOF-MS spectra were measured by a Bruker Autoflex Speed MALDI-TOF (Karlsruhe, Germany) using either α-cyano-4-hydroxycinnamic acid (CHCA) or *trans*-2-[3-(4-*tert*-butylphenyl)-2-methyl-2-propenylidene]malononitrile (DCTB) as matrix material. UV/VIS absorption spectra were measured on a JASCO V-650 spectrometer (de Meern, The Netherlands) with a JASCO CTU-100 Circulating Thermoset Unit at 20 °C. The compound of interest was dissolved in water (*c* = 0.4 mmol_catalyst_/L). Dynamic Light Scattering (DLS) spectra were obtained from a Malvern µV Zetasizer (Etten-Leur, The Netherlands) with a 830 nm laser and scatter angle of 90°. Disposable UV-transparent cuvettes from Sarstedt (Nuembrecht, Germany) with a path length of 10 × 2 mm were used in the DLS. The sample concentration was 1.0 mg/mL. Size exclusion chromatography (SEC) measurements of poly(pentafluorophenyl acrylate) were performed on a Shimadzu Prominence-I LC-2030C 3D (Den Bosch, The Netherlands) with a Shimadzu RID-20A refractive index detector. THF was used as elution solvent with a flow of 1.0 mL/min at an operating temperature of 40 °C. A mixed-C and mixed-D column in series (exclusion limit = 2,000,000 g/mol, 7.5 mm i.d. × 300 m), calibrated using polystyrene (Agilent, Santa Clara, CA, USA), was used to determine the molecular weight. SEC measurements of the functionalized polymers were performed on a PL-GPC-50 plus from Polymer Laboratories (Varian Inc. Company, Palo Alto, CA, USA) with a refractive index detector. DMF with 10 mM LiBr was used as elution solvent with a flow of 1.0 mL/min and at an operating temperature of 50 °C. The implemented column was a Shodex GPC-KD-804 column (exclusion limit = 400,000 Da, 0.8 cm i.d. × 300 mm), which has been calibrated with poly(ethylene oxide) (Polymer Laboratories). Dry solvents were obtained from MBraun solvent purification system (MB SPS-800, Stratham, NH, USA). To purify the functionalized polymers, standard RC Tubing from Spectrum Chemical (New Brunswick, NJ, USA) with a molecular weight cut-off (MWCO) of 6–8 kDa was used. All photocatalytic reactions were performed using 10 light-emitting diodes (LEDs) of Intelligent LED solution model ILH-XQ01-S380-SC211-WIR200 (385 nm, 3.85 W UV LEDs, Thatcham, UK). Automated column chromatography was performed on a Biotage Isolera^®^ One (Hengoed, UK) using a 150 g Biotage Silica cartridge.

### 4.3. General Procedure for Photoredox Dehalogenation in Water

A solution of 10 mg of functionalized polymer in 0.5 mL of demineralized water was prepared. To this solution, the substrate (50 µmol, 1.0 eq.), DIPEA (43.7 µL, 250 µmol, 5.0 eq.) and a magnetic stirrer were added. The sample was subsequently degassed for 5 min with argon. The top of the vial was wrapped with Parafilm to make the vial as airtight as possible. The degassed sample was placed in the irradiation set up and illuminated with 385 nm LED while being cooled with air for 3 or 18 h, depending on the experiment. After the catalytic reaction, the sample was extracted 3 times with 0.5 mL of CDCl_3_. Conversion was then measured by ^1^H NMR spectroscopy (estimated measurement error ± 2%).

*Synthesis of 10-(4-(2-((tert-butoxycarbonyl)amino)ethyl)phenyl)-3,6-di-tert-butyl-9-mesityl acridin-10-ium tetrafluoroborate* (**ACR-NH_2_**). The synthesis was adapted from the literature [35]. 3,6-Di-*tert*-butyl-9-mesitylxanthylium tetrafluoroborate (900 mg, 1.8 mmol, 1.0 eq.) was added to an oven dried flask and dry DCM (3.6 mL) was added. Subsequently, acetic acid (309.8 µL, 5.4 mmol, 3.0 eq.) and NEt_3_ (377.5 µL, 2.7 mmol, 1.5 eq.) were added to the solution. *tert*-Butyl (4-aminophenethyl)carbamate (853.4 mg, 2.6 mmol, 2 eq.) was dissolved separately in 2 mL of dry DCM and the solution was added dropwise to the reaction mixture. Next, the flask was wrapped in aluminum foil and the reaction mixture was stirred for 16 h at room temperature. The reaction mixture was subsequently washed with water (1 × 150 mL) and brine (1 × 50 mL). HBF_4_·Et_2_O-complex (220 µL, 1.8 mmol, 1.0 eq.) was added to the organic phase and swirled until homogeneity. The organic phase was washed with water (1 × 100 mL) and 1M aq. NaBF_4_ (1 × 100 mL) and afterwards dried over NaBF_4_. After removing the solvent, the solid compound was triturated 3x with 1:2 Et_2_O:pentane and the obtained product was dried for 18 h in vacuo at room temperature. 10-(4-(2-((*tert*-butoxycarbonyl)-amino)ethyl)phenyl)-3,6-di-*tert*-butyl-9-mesitylacridin-10-ium tetrafluoroborate (1.144 g, 1.60 mmol, 88% yield) was obtained as a dark yellow solid. ^1^H NMR (400 MHz, CDCl_3_): δ = 7.83 (t, *J* = 8.0 Hz, 2H), 7.79 (m, 4H), 7.61 (d, *J* = 8.0 Hz, 2H), 7.44 (s, 2H), 7.17 (s, 2H), 5.03 (br, 1H), 3.57 (q, *J* = 8.0 Hz, 2H), 3.14 (t, *J* = 7.4 Hz, 2H), 2.49 (s, 3H), 1.85 (s, 6H), 1.30 (s, 18H) ppm. ^13^C NMR (100 MHz, CDCl_3_): δ = 163.8, 162.3, 156.2, 143.9, 142.2, 140.3, 136.0, 134.8, 132.0, 129.2, 129.0, 128.3, 127.8, 127.5, 124.0, 115.1, 41.7, 36.8, 36.7, 36.1, 30.2, 28.5, 21.3, 20.2 ppm. ^19^F NMR (376 MHz, CDCl_3_: δ = −153.7 (m), −153.8 (m) ppm. MALDI-TOF *m*/*z* calculated for C_43_H_53_N_2_O_2_ [M-BF_4_]^+^ 629.41, found 629.43. 

10-(4-(2-((*tert*-Butoxycarbonyl)amino)ethyl)phenyl)-3,6-di-*tert*-butyl-9-mesitylacridin-10-ium tetrafluoroborate (250.0 mg, 348.8 µmol, 1.0 eq.) was dissolved in 4M HCl in dioxane (5 mL, 17.4 mmol, 50 eq.) and the solution was stirred at room temperature. After 6 h, total conversion was confirmed by TLC (heptane:ethanol 4:1) and 5.5 mL of 4M NaOH solution was added to the solution. The reaction mixture was stirred rigorously for 24 h at room temperature. Afterwards, the aqueous layer was extracted with DCM (4 × 25 mL) and the combined organic layers were washed with water (3 × 25 mL). 1M NaBF_4_ solution (100 mL) was added to the organic layer and stirred vigorously for 45 min. The organic layer was separated from the aqueous phase and dried over NaBF_4_. After removing the solvent under reduced pressure, the solid compound was triturated 3x with 1:2 Et_2_O:pentane and dried for 18 h in vacuo at room temperature. 10-(4-(2-Aminoethyl)phenyl)-3,6-di-*tert*-butyl-9-mesitylacridin-10-ium tetrafluoroborate **ACR-NH_2_** (193.46 mg, 313.76 µmol, 90% yield) was obtained as bright yellow solid. ^1^H NMR (400 MHz, CDCl_3_): δ = 8.01 (d, *J* = 8.1 Hz, 2H), 7.78 (d, *J* = 1.9 Hz, 4H), 7.63–7.54 (m, 2H), 7.47 (s, 2H), 7.16 (s, 2H), 6.88 (s, 3H), 3.72 (d, *J* = 10.5 Hz, 2H), 3.58–3.49 (m, 2H), 2.48 (s, 3H), 1.83 (s, 5H), 1.31 (s, 18H) ppm. ^13^C NMR (100 MHz, CDCl_3_): δ = 163.3, 161.3, 141.1, 140.5, 139.3, 134.9, 134.2, 131.7, 128.2, 128.0, 127.2, 126.9, 126.7, 122.9, 114.2, 40.6, 35.8, 31.9, 29.2, 21.3, 20.3 ppm. ^19^F NMR (376 MHz, CDCl_3_): δ = −150.8 (d, *J* = 19.6 Hz) ppm. MALDI-TOF *m*/*z* calculated for C_38_H_45_N_2_ [M-BF_4_]^+^ 529.36 found 529.37.

## 5. Conclusions

In this work, two reductive organic photocatalysts, namely 10-phenylphenothiazine and 9-mesityl-3,6-di-*tert*-butyl-10-phenyl-acridinium, were grafted to a polymeric backbone. The additional presence of hydrophobic and/or hydrophilic grafts, *n*-dodecylamine and Jeffamine, induced a hydrophobic collapse of the polymer in aqueous media, resulting in the formation of polymeric nanoparticles with hydrodynamic diameters D*_H_* of 11 nm and smaller. Owing to the customizability of the synthetic approach, six different types of amphiphilic polymers, each differing in either polymer backbone length or composition of functionalized groups, were readily accessible. These **PNs** were constructed in such a way that the effect of polymer microstructure—polymer length, catalyst loading and catalyst type—on catalyst activity and selectivity could be compared to one another.

All **PNs** were successful in the reduction of aryl halides in an aqueous medium upon irradiation with light of 385 nm for 3 h. Due to the hydrophobic compartmentalization of these **PNs**, a high local substrate and catalyst concentration is obtained, resulting in shorter reaction times. As expected, phenothiazine, the less powerful but faster catalyst, showed higher conversions compared to acridinium (at equal catalyst loading) for substrates that had a lower reduction potential. The catalytic activity of the phenothiazine, however, dropped quickly when more resilient aryl halides were used as substrates. The **PNs** with a higher catalyst loading, which had no additional *n*-dodecyl grafts, showed the highest conversion for most substrates. Altering the length of the polymeric backbone did not show any clear difference in catalytic activity. 

Finally, the **PNs** were used to perform a C-C cross-coupling reaction between 4-chlorobenzonitrile and *N*-methylpyrrole that competes with the corresponding reduction reaction. Due to the high local concentration of the substrate within the nanoparticle, shorter reaction times and significantly less *N*-methylpyrrole were necessary to achieve reasonable selectivity towards the C-C cross-coupled product. The results show that attaching organic photocatalysts to amphiphilic polymers is an effective way to create tailor-made nanoreactors for efficient catalysis in water. By using these **PNs**, shorter reaction times and less substrate are needed to reach a high conversion. This, together with the fact that these photocatalysts are light driven, makes the use of these **PNs** an effective and sustainable approach to perform photoredox catalysis in aqueous media.

## Data Availability

All data are available from the corresponding author upon request.

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
