# Peer review of "Photoredox-Catalyzed Reduction of Halogenated Arenes in Water by Amphiphilic Polymeric Nanoparticles"

_molecules, 2021, doi:10.3390/molecules26195882_

Round 1

Reviewer 1 Report

This article is devoted to the development of the concept of conducting water-media reactions in nanoreactors formed in situ. The manuscript describes the application of this approach to the reductive hydrodechlorination of chloroaromatics under the action of light using organic photocatalysts. This is a hot topic.
The study is well planned and well executed.
The only comment the manuscript elicits is the lack of discussion of the reaction mechanism. I believe that the authors should, based on a large amount of reliable literature data, suggest a possible reaction mechanism and depict its scheme in the manuscript (indicating, among other things, what is the source of the hydrogen atom, that is, the reducing agent). It seems to me that under these conditions, in the absence of recombination products or disproportionation of aryl radicals (which should be reliably proven), such a reducing agent is DIPEA, but this is my personal opinion, which I do not insist on. In any case, it is necessary to show where the hydrogen atom comes from and what is formed as a result of the reaction (not only arene, but also the salt, as well as the oxidation product of the reducing agent).
In general, I can recommend the manuscript for publication, taking into account the remark made.

Author Response

We are thankful for the positive evaluation of our manuscript and the constructive feedback. Indeed, the mechanism for the light-induced dehalogenation of arenes has been investigated and described in detail by us and in other literature reports. The cleavage of the carbon-halogen bond is induced by a single-electron transfer from the excited photoredox catalyst to the halogenated arene. As a result, a halogen anion is released and a benzene radical is formed. Subsequently, proton abstraction from the tertiary amine base, in this case DIPEA, leads to the formation of the reduced benzene as the final product and the DIPEA halide salt as a side product. Studies with deuterated solvents as well as amine bases showed that the base is the primary proton source (Chem. Commun., 2015, 51, 11705; Chem. Eur. J. 2020, 26, 10355). In addition, the amine base acts not only as a proton source but also as an electron source to regenerate the PTH catalyst after the single-electron transfer and to reduce the ACR catalyst in its excited state. As the mechanisms have been reported in literature before, we decided to add two schemes describing the mechanisms to the updated Supporting Information (Figures S12 and S13). In addition, we described the role of the tertiary amine base DIPEA in the catalytic cycle in the main text.

Reviewer 2 Report

In this work, the authors evaluated six different amphiphilic polymeric nanoparticles to assess the effect of polymer length, catalyst loading and nature of the catalyst (PTH or ACR) in the dechlorination of a range of aromatic chlorides. The context is novel and interesting. I believe this work to be of immediate interest to a broad range of researchers in the material and chemical science, and is therefore suitable for publication in Molecules.

  • Authors should carefully modify the logic of the paper. For example, what does "hydrophobic domains present within these nanoparticles after hydrophobic collapse" in the introduction mean?
  • This article is a scientific paper. Many descriptions should apply specific data to state, for example:"polymeric nanoparticles of nanometer-size" What size? "The local catalyst and substrate concentrations are high" What concentration?
  • Some related and cutting-edge work about confinement effect should be cited in the article: 10.1021/acsami.0c13802, 10.3390/molecules26175292, 10.1002/adma.201908243, 10.3390/molecules25143311, 10.1021/acsami.7b10540

Author Response

In this work, the authors evaluated six different amphiphilic polymeric nanoparticles to assess the effect of polymer length, catalyst loading and nature of the catalyst (PTH or ACR) in the dechlorination of a range of aromatic chlorides. The context is novel and interesting. I believe this work to be of immediate interest to a broad range of researchers in the material and chemical science, and is therefore suitable for publication in Molecules.

Authors should carefully modify the logic of the paper. For example, what does "hydrophobic domains present within these nanoparticles after hydrophobic collapse" in the introduction mean?

This article is a scientific paper. Many descriptions should apply specific data to state, for example:"polymeric nanoparticles of nanometer-size" What size? "The local catalyst and substrate concentrations are high" What concentration?

We thank the Reviewer for critically evaluating our manuscript. We changed the manuscript according to the comments of the Reviewer and thus made a few sentences in the abstract clearer. For instance, we specified the size of the nanoparticles by referring to their hydrodynamic diameter and altered the logic of the sentence in lines 15-18. The local catalyst concentration within the nanoparticles can be estimated by calculating the volume of the sphere-like nanoparticles based on the hydrodynamic diameter and the mole of covalently bound catalysts per nanoparticle. For instance, the local concentration of PTH catalysts in single-chain nanoparticles of PN1 is approximately 3.6 mol/L, while the overall concentration of PTH in aqueous solution under the applied conditions is 1.1 mmol/L (three orders of magnitude lower). Estimating the local concentration of aromatic substrates within the nanoparticles is challenging as the partition coefficient (log P value), which describes the ratio of substrate concentration between the water phase and the inner compartments of the nanoparticles, is not known. In addition, the equilibrium between the two phases is also dependent on the nature of each substrate, which overall makes a general statement regarding the local concentration of the substrates within the nanoparticles difficult. As a discussion about the local concentration is associated with a large error, we decided to not further discuss this aspect in the main text.

Some related and cutting-edge work about confinement effect should be cited in the article: 10.1021/acsami.0c13802, 10.3390/molecules26175292, 10.1002/adma.201908243, 10.3390/molecules25143311, 10.1021/acsami.7b10540

For adding the suggested references, we do not see any relation with the work described here, so we respectfully decline to add these.

Reviewer 3 Report

The manuscript of Palmans A.R.A. and coworkers is interesting and can be published in Molecules. However, it needs some minor revision. I propose to consider the following points in the revised version:

1) Title of the manuscript should be the same in the main text and in the supplementary materials.

2) Line 22 (abstract) can be “ACR … more effective than PTH.”

3) In the whole manuscript, a space should be included between the text word and the cited literature, e.g., “compounds [1-5].”

4) All abbreviations should be explained where first used in the text.

5) Chapter 2 describes methodology and results. Hence, a more appropriate title should be given, e.g., “Methodology and Results”. Other possibility is “Results and Discussion”.

6) In experiments, Authors applied chlorobenzenes substituted by an electron donor (Me, OMe) or electron acceptor (CF3, COOMe, CN) group R (Table 1). The R group affects by only inductive (Me and CF3) and by both various inductive and resonance effect (OMe, COOMe, CN). Some explanation for the choice of substituents should be included in the text, and also some explanation why strong electron donor NMe2 or N=C(NMe2)2 and strong electron acceptor NO2 were not considered.

7) The terms “electron poor aromatic systems” and “electron rich systems” and also the type of measure of aromatic electrons should be explained.

8) For Table 2, I propose the following abbreviations: R00 (benzene), R0 (chlorobenzene), R6 (bromobenzene), R7 (iodobenzene).

9) For Table 3, N-methylpyrrole can be abbreviated as S8 and product abbreviated as “R11” can be named “R8”.

10) Authors should add some comments on reliability of experimental methods used for analysis of reaction products.

11) Chapters “Discussion” and “Conclusions” should be revised. Chapter Conclusions cannot repeat Abstract, and conclusions cannot be included in chapter Discussion.

12) After ACR-NH2 p.11 in Table of Contents (SM), PTH-NH2 p. 12 should be included.

Author Response

The manuscript of Palmans A.R.A. and coworkers is interesting and can be published in Molecules. However, it needs some minor revision. I propose to consider the following points in the revised version:

We are grateful for the constructive assessment of our manuscript of the Reviewer and we altered the manuscript in accordance with the Reviewer comments.

1) Title of the manuscript should be the same in the main text and in the supplementary materials.

The title in the Supporting Information was changed to the title of the manuscript.

2) Line 22 (abstract) can be “ACR … more effective than PTH.”

The sentence in the abstract (line 22) was edited as suggested.

3) In the whole manuscript, a space should be included between the text word and the cited literature, e.g., “compounds [1-5].”

Spaces before each reference were added in the entire manuscript.

4) All abbreviations should be explained where first used in the text.

We introduced the abbreviations for size-exclusion chromatography (SEC), N,N-diisopropylethylamine (DIPEA), and light-emitting diode (LED, line 154) properly.

5) Chapter 2 describes methodology and results. Hence, a more appropriate title should be given, e.g., “Methodology and Results”. Other possibility is “Results and Discussion”.

The title of the second chapter was changed to “Methodology and Results”.

6) In experiments, Authors applied chlorobenzenes substituted by an electron donor (Me, OMe) or electron acceptor (CF3, COOMe, CN) group R (Table 1). The R group affects by only inductive (Me and CF3) and by both various inductive and resonance effect (OMe, COOMe, CN). Some explanation for the choice of substituents should be included in the text, and also some explanation why strong electron donor NMe2 or N=C(NMe2)2 and strong electron acceptor NO2 were not considered.

Regarding the choice of substrates, we did not include substrates with strong electron donors such as NMe2 as they could potentially act as the proton or electron source according to the reaction mechanisms that is illustrated in newly added Figures S12 and S13 and thus result in side reactions. Nitro compounds as electron poor substrates were initially tested but yielded many side products as nitro substrates undergo photolysis reactions under light illumination (Environ. Sci. Technol. Lett. 2021, 8, 747). We included that discussion in the main text.

7) The terms “electron poor aromatic systems” and “electron rich systems” and also the type of measure of aromatic electrons should be explained.

The reactivity of aromatic compounds is strongly influenced by electronically active substituents, which is also observed in this study. The Hammett constants of these substituents give a valuable indication of how strong the influence would be and we used these constants for our substrate selection. Yet, we did not further specify these terms in the main text because we expect that the reader of this journal has generally a good understanding of the electronic properties of aromatic systems.

8) For Table 2, I propose the following abbreviations: R00 (benzene), R0 (chlorobenzene), R6 (bromobenzene), R7 (iodobenzene).

9) For Table 3, N-methylpyrrole can be abbreviated as S8 and product abbreviated as “R11” can be named “R8”.

The Reviewer suggested to give some of the products and starting materials different abbreviations. However, we respectfully decline these suggestions as these abbreviations follow a correct order which is in line with the compounds mentioned in the Supporting Information. For instance, N-methylpyrrole cannot be named S8 as the compound 2-bromo-4-chloro-1-iodobenzene is already abbreviated with S8.

10) Authors should add some comments on reliability of experimental methods used for analysis of reaction products.

In order to analyze the product formation after the indicated reaction time, we primarily relied on NMR spectroscopy and analyzed the characteristic proton signals in the aromatic region. Although the accuracy of NMR spectroscopy is high, we estimate the measurement error to be in the range of ± 2%. We added this comment to the general procedure in the main text and the Supporting Information.

11) Chapters “Discussion” and “Conclusions” should be revised. Chapter Conclusions cannot repeat Abstract, and conclusions cannot be included in chapter Discussion.

We have removed a redundant part in the discussion, as well as in the conclusion section.

12) After ACR-NH2 p.11 in Table of Contents (SM), PTH-NH2 p. 12 should be included.

The Table of Contents in the Supporting Information was updated accordingly.